# Sociodemographic Associations and COVID-19 Symptoms Following One Year of Molecular Screening for SARS-CoV-2 Among Healthcare Workers

**DOI:** 10.3390/v17121622

**Published:** 2025-12-16

**Authors:** Viviane Campos Barbosa de Sena, Michelle Oliveira, Rejane Alencar Saldanha, Larissa Vicenza, Tais Hanae Kasai Brunswick, Bernardo Tura, Helena Cramer Veiga Rey, Adriana Bastos Carvalho, Antônio Carlos Campos de Carvalho, Djane Braz Duarte, Dayde Lane Mendonça da Silva, Daniel Arthur Barata Kasal

**Affiliations:** 1Department of Research and Education, Ministry of Health, National Institute of Cardiology, Rio de Janeiro 22240-006, RJ, Brazil; vivianecbsena@gmail.com (V.C.B.d.S.); carvalhoab@biof.ufrj.br (A.B.C.); acarlos@biof.ufrj.br (A.C.C.d.C.); 2Centro Nacional de Biologia Estrutural e Bioimagem, Universidade Federal do Rio de Janeiro, Rio de Janeiro 21941-902, RJ, Brazil; 3Centro de Pesquisa de Medicina de Precisão, Universidade Federal do Rio de Janeiro, Rio de Janeiro 21941-599, RJ, Brazil; 4Pharmacy Department, School of Health Sciences, University of Brasília, Brasília 70910-900, DF, Brazil; 5Internal Medicine Departament, State University of Rio de Janeiro, Rio de Janeiro 21040-360, RJ, Brazil

**Keywords:** COVID-19, vaccine, healthcare professionals, RT-PCR

## Abstract

Background: During the COVID-19 pandemic, high rates of infection with SARS-CoV-2 were reported in healthcare workers (HCWs), among whom asymptomatic individuals had high potential to spread the virus while assisting high-risk patients. This study conducted routine SARS-CoV-2 screening among the staff of a specialized cardiology hospital in Brazil during 2022 and 2023, while also evaluating variables associated with infection and the occurrence of symptoms. Methods: A prospective cohort study of 94 HCWs with biweekly RT-PCR screening was performed, employing RT-PCR from nasal swabs. Results: Participants aged 50.9 ± 10.2 years and were predominantly female (85.1%) and non-white (56.4%). The follow-up period was 576.4 ± 185.9 days, and most participants worked in the intensive care unit/emergency department (34%). Although the HCWs with the highest COVID-19 rates before inclusion were technicians/graduates (67.3%) and non-white individuals (57.7%), these groups presented lower infection rates at follow-up (*p* < 0.001, CI 95% 2.924–27.93; and *p* = 0.02, CI 95% 0.129–0.859, respectively). The number of asymptomatic cases increased during the study (*p* = 0.001), and simultaneous infection upsurges occurred in different hospital departments. Interpretation: These data highlight the association between educational level and the risk of SARS-CoV-2 infection in HCWs. The synchronicity of cases in different hospital departments offers insights about the nosocomial spread of SARS-CoV-2. The increase in the number of asymptomatic infections with repeated infections suggests that regular molecular screening may contribute to increasing the safety of both patients and HCWs in a pandemic context.

## 1. Introduction

In December 2019, a highly lethal acute respiratory syndrome was described in the city of Wuhan, Hubei province, China, initially displaying an epidemiological association with the seafood and wildlife market [1]. In less than a month, the causal agent was characterized as Severe Acute Respiratory Syndrome Coronavirus-2 (SARS-CoV-2), and the associated disease was named coronavirus disease-19 (COVID-19). Since the earliest reports, there has been persistent evidence of increased exposure and high rates of infection among healthcare workers (HCWs) [2]. Accordingly, molecular surveillance for SARS-CoV-2 in HCWs has been suggested as a tool for assessing vaccine effectiveness and mitigating nosocomial transmission of the infection [3]. Most previous studies assessing SARS-CoV-2 infection among HCWs were retrospective and relied primarily on serological data [4,5]. The pivotal molecular-based evaluation published to date is the SIREN study, which conducted biweekly molecular testing of more than 8000 HCWs in England, for 12 months [6]. Further investigations involving HCWs from diverse geographic regions could strengthen our understanding of the epidemiological characteristics of COVID-19 within this professional group.

The advent of COVID-19 vaccines represented a major breakthrough for pandemic control, reducing mortality and disease severity. Indeed, approximately half of the cases of SARS-CoV-2 infection in vaccinated individuals are asymptomatic [6]. However, asymptomatic infections still have the potential to transmit the virus through direct contact [7]. This study aimed to identify, through regular molecular screening, the occurrence of SARS-CoV-2 infection in HCWs working in a specialized cardiology hospital linked to the Brazilian Public Healthcare System.

## 2. Materials and Methods

### 2.1. Study Design and Participants

This was a prospective cohort study with adult HCWs actively working in the National Institute of Cardiology, a public, quaternary-care hospital in the city of Rio de Janeiro, Brazil. Volunteer recruitment was performed through advertising on the hospital website and in common areas, administering invitations through institutional emails, and through direct engagement by the research team. The total number of invited HCWs was estimated to be 1820. Eligibly criteria included HCWs with a technical or higher education who were available to be monitored during the follow-up period through visits, telephone contact, or other means of digital communication. The exclusion criterion was having participated in clinical trials for the development of COVID-19 vaccines. The Informed Consent Form (ICF) was administered during the first visit. This study was approved by the Local Ethics Research Committee under protocol #CAAE 53201021.0.1001.5272.

### 2.2. Clinical Data Collection

Clinical and sociodemographic information was obtained through an interview at the time of recruitment. Before collecting each swab, the project’s nursing team performed an anamnesis directed to COVID-19 symptoms. Symptom assessment was based on participant recall, consistent with the approach employed by previous studies [6,8]. Symptomatic HCWs were not followed up until the next test collection. COVID-19 symptoms during the study included self-reported fever, dyspnea, palpitations, taste disturbances, headache, nose congestion, cough, sore throat, and malaise. Screening for SARS-CoV-2 was performed every 15 days through molecular testing to detect the virus using the quantitative real-time polymerase chain reaction (RT-PCR) with material obtained via nasal swabs. The biweekly PCR screening interval was selected based on prior literature [6] and systematic reviews assessing the duration of SARS-CoV-2 shedding in biological specimens, which generally spans from approximately four days before to 17 days after the onset of symptoms [9,10].

### 2.3. Sample Collection

Nasopharyngeal secretions were collected using a sterile Rayon swab by a professional wearing personal protective equipment (PPE). After collection, the swabs were placed in a conical tube containing 2 mL of viral transport medium and were transported in a refrigerated container in accordance with international standards (UN 3373, category B) [11]. The samples were stored at the hospital at −80 °C. Specimen handling for RT-PCR was performed in certified biosafety cabinet by trained personnel wearing PPE. All used materials were disposed of in properly designated biohazard waste containers.

### 2.4. Laboratory Diagnosis of SARS-CoV-2 Infection

RT-PCR was performed in a single step using GoTaq Probe 1-Step RT-qPCR (Promega, Madison, WI, USA) or TaqPath 1-Step RT-qPCR Master Mix (Thermo Scientific, Waltham, MA, USA). Detection of the RNA of interest was performed using a TaqMan system containing two oligonucleotides and a probe. The primer sequences used were provided by the United States Centers for Disease Control and Prevention (CDC) and are commercialized by Integrated DNA Technologies (Coralville, IA, USA). The assay was performed and analyzed according to the CDC instructions (available at: https://www.fda.gov/media/134922/download (accessed on 23 August 2025)), and the quantitative PCR equipment used was QuantStudio 5 (Thermo Scientific, MA, USA), while the analysis software was QuantStudio Design and Analysis v1.4.x. Reinfection during the study was defined as the detection of SARS-CoV-2 in a participant who had a prior documented infection—that is, a new positive RT-PCR result preceded by a negative result, with an earlier positive molecular diagnosis recorded in the study database.

### 2.5. Employee Flow After Screening

Individuals were informed of their test results by the institution’s Occupational Health Service (OHS). The OHS was also responsible for providing guidance regarding home isolation, with instructions for the employees and their family members.

### 2.6. Statistical Analysis

Data are presented as absolute and relative frequencies, the mean and standard deviation, or the median and distance, according to their characteristics. Contingency tables were constructed to assess the associations between discrete variables and outcomes. Numerical variables were analyzed using the tests indicated according to the type of variable studied. Continuous and categorical variables are presented as the median with the interquartile range (IQR) and n (%), respectively. Normality of data was assessed employing the Shapiro–Wilk test. We used the Mann–Whitney U test, χ^2^ test, or Fisher’s exact test to compare differences between COVID-19-positive and -negative individuals, when appropriate. Odds ratio (OR) and confidence intervals were calculated to measure the strength of association between an exposure and a COVID-19 diagnosis. Binary logistic regression was used to explore the relationship between COVID-19 diagnosis and participant age. Data were analyzed using Jamovi 2.7.9 and R 4.5.1 statistical software. Participants with missing data were to be excluded from analysis, which was not the case in our study.

## 3. Results

Between 9 February 2022 and 22 December 2023, regular molecular testing was performed on 94 participating HCWs. Over the course of the project, 13 (13.8%) of the participants spontaneously withdrew from the study or were absent for a certain period due to medical leave. However, their regular testing data was kept until their leave from the study.

The COVID-19 vaccines available in Brazil during this period were Coronavac^®^, AstraZenica^®^, Janssen^®^, Pfizer^®^, bivalent Pfizer^®^ and Sinovac^®^. The average number of COVID-19 vaccine doses administered to individuals before study inclusion was 4.2 ± 0.85. The time interval between the last dose of the COVID-19 vaccine and the first infection in asymptomatic individuals was 168.8 ± 71.9 days and in symptomatic individuals, 168.5 ± 107.5 days.

### 3.1. Sociodemographic Profile of Participants

Table 1 presents the sociodemographic variables of the participants. The majority were women (85.1%), and the mean age was 50.9 ± 10.2 years. Forty-one (43.6%) declared themselves as white, while 53 (56.4%) declared themselves as non-white. Regarding education, 70 (74.5%) were health technicians or had graduate degrees, while 24 (25.5%) had a specialization, master’s degree, or doctorate. The distribution of HCWs among departments was as follows: 32 (34%) worked in the intensive care unit (ICU)/emergency department, 24 (25.5%) worked on wards, 13 (13.8%) worked in the laboratory/radiology, and 25 (26.6%) worked in other sectors (rehabilitation, surgical center, material and sterilization center, or continuing education). The most common comorbidities among HCWs were systemic arterial hypertension (18.1%), diabetes (16%), and asthma (9.6%). The number of household contacts of the participating HCWs was 2.9 ± 1.3. This study’s follow-up period was 576.5 ± 185.9 days, and the number of swabs collected per HCW was 22.7 ± 11.1.

### 3.2. COVID-19 Before Study Inclusion and During Follow-Up

Previous COVID-19 infection was reported by 55.3% of participants, and the average interval between the last infection and study inclusion was 494.5 ± 277.7 days. In Table 2, individuals with and without a previous COVID-19 infection are compared regarding gender, age, ethnicity and education. No significant differences were observed between the two groups, although higher previous COVID-19 infection rates were observed in technicians/graduates (67.3%) and in those of a non-white ethnicity (57.7%).

### 3.3. Comparison Between HCWs Without and with One SARS-CoV-2 Infection During Follow-Up

During follow-up, 28 individuals (29.8%) presented with only one infection, while 53 participants (56.4%) were not infected by SARS-CoV-2. Higher positivity rates were observed among HCWs with a higher education level (*p* < 0.001) and a white ethnicity (*p* = 0.02). Working in the ICU/emergency department was also associated with higher infection rates during the study (*p* = 0.04). These data are presented in Table 3.

We performed binomial logistic regression, in order to evaluate the effects of a potential confounding factor (age), both for COVID-19 infection previous to study inclusion and during the study (Appendix A). However, no association with the infection was found.

### 3.4. Symptom Occurrence at First and Second Infection During Follow-Up

The occurrence of symptoms was reduced with repeated infections during follow-up. The absence of symptoms was observed in 24.4% of HCWs during the first infection, while it increased to 92.3% with the second infection (*p* = 0.001, Table 4). No third episodes of infection were detected. In all instances of asymptomatic positive participants throughout the study, no symptom was reported by the next biweekly testing.

### 3.5. SARS-CoV-2 Detection Among Hospital Departments During Follow-Up

During this study, three peaks of infection were observed in the participating HCWs (in June and November 2022 and October 2023). For analysis purposes, departments were designated as Department A, Department B, Department C, and others. Figure 1 presents the progression of SARS-CoV-2 over time, displaying synchronous peaks of case detection in different hospital departments.

## 4. Discussion

With the ongoing threat posed by COVID-19 to healthcare systems and society during the pandemic period, several policies to prevent and reduce the spread of SARS-CoV-2 were developed. These strategies, including quarantine, social isolation, and total lockdowns, were introduced in many countries to contain the spread of the virus. Although certain population groups, such as ethnic minorities and people with pre-existing chronic conditions, were disproportionately affected by COVID-19, HCWs were at higher risk of contracting COVID-19 compared with the general population [12]. HCWs were also at higher risk of physical and psychological harm due to extreme workload pressure, isolation from family members, and indirect complications of COVID-19, such as protective-equipment-related injuries (rash, dermatitis, pressure ulcers, etc.) [12,13].

HCWs who were on the frontlines of the COVID-19 response were at higher risk of acquiring the disease and subsequently exposing patients, other HCWs, and family members to it. A meta-analysis of eleven studies showed that almost 10% of COVID-19-positive individuals were HCWs. Notwithstanding this, the incidence of severe disease and mortality among HCWs was lower (9.9%) when compared to all COVID-19-positive patients [14]. There were no COVID-19-related hospitalizations or deaths in our study.

Similarly to our study, North American data indicate that ethnic minorities disproportionately worked in the top nine occupations most exposed to COVID-19, therefore meaning they were at high risk of infection. Importantly, 40% of HCWs self-identified as racial minorities, including 16% as Black and 13% as Hispanic [15]. In our study, the participating HCWs were predominantly female and self-identified as non-white. These data reflect the typical composition of the healthcare workforce in Brazil regarding nursing and direct patient care professionals [16].

COVID-19 infection before study inclusion was reported by most participants in our study, a higher proportion when compared to a recent meta-analysis performed on HCWs infection rates [17]. Our finding suggests that most professionals already had some degree of acquired immunity, either through natural infection or vaccination.

Among the study participants reinfected, only one presented symptoms at the time of swab collection. Previous studies have shown that reinfection was associated with faster viral clearance than the first infection, likely due to the presence of immunological memory, suggesting that reinfection is an uncommon but possible event in individuals continuously exposed to SARS-CoV-2, such as HCWs. Generally, individuals with reinfection develop mild disease without significant complications [12].

In our study, the occurrence of infections and reinfections, even after vaccination, may be associated with factors such as the circulation of new immune-evasive variants [18] and the decline in immunity over time. Moreover, the immune response may vary between individuals, and factors such as age, comorbidities, and interval between doses may influence the protection provided by the vaccine [6].

Our study observed a distribution of infected professionals in different hospital departments. This dispersion allows us to speculate that reinfection occurred in different hospital areas and may even have occurred in common areas (not investigated by the authors). Ettorre et al. demonstrated that infected HCWs mainly worked on general wards (77.5%), followed by in the emergency department (17.5%) and intensive care (5%) [19].

Our findings reinforce the importance of vaccination in mitigating the impacts of COVID-19, especially among HCWs. Furthermore, they highlight the need for continuous surveillance, especially in professionals subject to greater exposure, to identify patterns of reinfection and assess the duration of immunity conferred by natural infection and vaccination.

When comparing HCWs who had one molecular diagnosis of COVID-19 during follow-up with those who remained negative, we observed significant differences in some sociodemographic and clinical variables. A white ethnicity was associated with lower COVID-19 infection rates before study entry and with a higher infection rate during the follow-up period, a finding that may be related to occupational and socioeconomic factors or even differences in exposure to the virus within the hospital environment. Education was also shown to be a relevant factor, with those with a lower educational level displaying a lower infection rate during follow-up.

Worldwide, gender and ethnicity are associated with the educational level in healthcare workforce, with higher proportion of women and non-white individuals historically working in lower-education activities [20]. Accordingly, the effects of ethnicity or educational levels per se in SARS-CoV-2 infections cannot be separated in our study. HCWs with different educational levels may perform different functions within the health institution, which may influence their exposure to the virus [21]. HCWs with lower levels of education may be more frequently involved in operational functions that require greater physical contact with patients, increasing the risk of infection. Arguably, this group of professionals faced greater exposure to the virus during the first year of the pandemic. This, along with vaccination, could have provided a robust immune response, therefore decreasing SARS-CoV-2 infection rates in the later timeframe of our study (the second and third years of the pandemic).

The temporal analysis of COVID-19 cases among HCWs participating in this study revealed three periods of higher incidence. These peaks in positivity may be associated with the predominant circulation of different SARS-CoV-2 lineages, as well as seasonal and institutional epidemiological factors.

Among asymptomatic carriers and individuals at risk due to the asymptomatic transmission of COVID-19, HCWs represent an important but understudied population. Several studies have shown that asymptomatic carriers contribute substantially to the spread of the virus, even simply by breathing in a room [14]. HCWs may be at increased risk of SARS-CoV-2 infection due to close contact with highly infectious patients, but also due to exposure to undiagnosed or subclinical infectious cases. It is important to emphasize that throughout the pandemic period, the availability of and access to protective equipment for HCWs remained high, along with training and supervision by infection prevention and control committee members, with adequate adherence to protocols.

COVID-19 will probably continue to be a global health concern in years to come, with the emergence of new variants [22]. The high-risk groups for COVID-19 remain the same, and it is essential to continue taking preventive measures, such as vaccination and wearing masks in large gatherings.

The findings of the present study reinforce the importance of considering individual and occupational characteristics and comorbidities when assessing the risk of SARS-CoV-2 infection in HCWs. In addition, they highlight the need for specific protection strategies for more vulnerable groups, taking into account factors such as the function performed, comorbidities, and exposure in the work environment. Our data suggest that active screening protocols, including periodic molecular testing, can be incorporated into hospital epidemiological surveillance as an additional infection control strategy. This approach may be particularly relevant during periods of high viral circulation or hospital outbreaks.

## 5. Study Limitations

Our study had a smaller-than-expected number of volunteers (n = 94). The sample size constrained the analysis of variables using logistic regression, which in this study was limited to participant age. We believe the low adherence could be related to discomfort with the repeated testing and the reduction in infection rates in the population over time, which may have decreased volunteer interest in testing. Furthermore, staffing pressures or burn-out observed during the pandemic [23] could make it difficult for HCWs to attend frequent testing appointments, or commit to a year-long testing schedule.

In addition, individuals with increased contact with COVID-19 patients could display increased willingness to participate, whereas professionals who were ill and on leave would not have responded to recruitment efforts. These circumstances could introduce heterogeneity in exposure and selection bias. As a consequence, the study participants do not necessarily represent the universe of HCWs at the hospital. Additional infection-related variables—such as variant identification, and serological data—were not available in our study, which restricts the scope of its applicability.

This study was conducted at a single center, limiting the extrapolation of the present results for other healthcare facilities. Other limitations of our study include the lack of daily monitoring of symptomatic HCWs. Notwithstanding, it offered a picture of the practice in the setting of the COVID-19 pandemic at a public, specialized Cardiology healthcare institution.

## 6. Conclusions

The information obtained in the current study suggests the potential benefits of a strategy based on the structured genomic surveillance of professional staff at quaternary hospitals. Although the most severe impacts of the COVID-19 pandemic have been mitigated with the advancement of vaccination and the strengthening of the health system, SARS-CoV-2 still circulates in society, as do other viruses of epidemiological importance, such as influenza and respiratory syncytial virus. This scenario shows that the threat of new health emergencies remains latent and requires continued attention from authorities and society.

Periodic molecular screening in HCWs can help to characterize the effectiveness of the COVID-19 vaccine in this professional group. In addition, there is potential to reduce nosocomial transmission, considering cases in which the positive result occurred before the onset of symptoms.

## Figures and Tables

**Figure 1 viruses-17-01622-f001:**
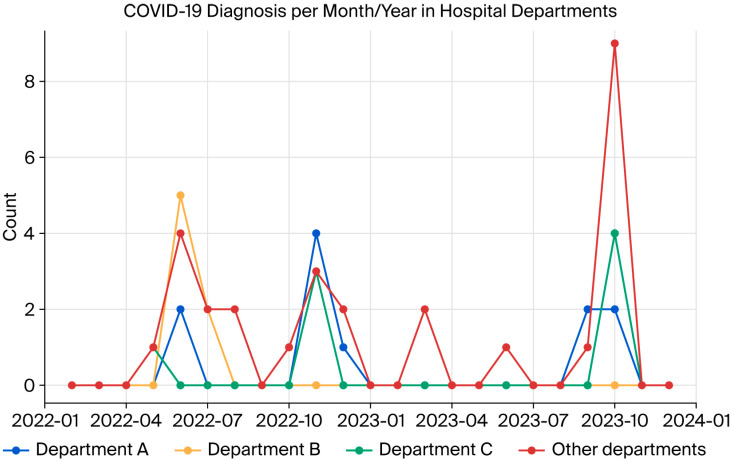
Dynamics of SARS-CoV-2 infection in hospital departments during the study.

**Table 1 viruses-17-01622-t001:** Participants’ sociodemographic profiles.

Variable	N (%)
Female	80 (85.1)
Age, mean ± SD (years)	50.9 ± 10.2
Self-declared ethnicity	
White	41 (43.6)
Non-white	53 (56.4)
Education	
Technician/graduate	70 (74.5)
Specialty/Master/Doctorate	24 (25.5)
Hospital department	
ICU/Emergency	32 (34)
Ward	24 (25.5)
Laboratory/Radiology	13 (13.8)
Other	25 (26.6)
BMI > 30 kg/m^2^	39 (41.5)
Comorbidities	
Hypertension	17 (18.1)
Diabetes	15 (16)
Asthma	9 (9.6)
Other	46 (48.9)
Household contacts, mean ± SD	2.9 ± 1.3

BMI, body mass index; ICU, intensive care unit.

**Table 2 viruses-17-01622-t002:** Comparison between HCWs with and without a previous COVID-19 infection.

Variable	Previous COVID-19				
	No	Yes	*p*-Value	OR	CI 95%
Female gender n (%)	35 (83.3%)	45 (86.5%)	0.66	0.77	0.25–2.42
Age (mean ± SD, years)	51.1 ± 11.9	50.8 ± 8.7	0.86	0.99	47.44 ± 54.83 *
					48.34 ± 53.23 **
Ethnicity n (%)					
White	19 (45.2%)	22 (42.3%)	0.77	1.12	0.49–2.55
Non-white	23 (54.8%)	30 (57.7%)			
Education n (%)					
Technician/Graduate	35 (83.3%)	35 (67.3%)	0.07	2.42	0.89–6.58
Specialization/Master/Doctorate	7 (16.7%)	17 (32.7%)			
Hospital department n (%)					
ICU/emergency	15 (35.7%)	17 (32.7%)	0.96	***	
Ward	11 (26.2%)	13 (25%)			
Laboratory/Radiology	5 (11.9%)	8 (15.4%)			
Other	11 (26.2%)	14 (26.9%)			
BMI, mean ± SD (kg/m^2^)	28.4 ± 4.8	29.1 ± 5.1	0.50	1.03	26.92 ± 29.94 *
					27.68 ± 30.57 **
Comorbidities n (%)					
Obesity	17 (40.5%)	22 (42.3%)	0.85	1.07	0.47–2.46
High blood pressure	8 (19%)	9 (17.3%)	0.82	0.89	0.31–2.55
Diabetes	7 (16.7%)	8 (14.4%)	0.86	0.90	0.30–2.75
Asthma	4 (9.5%)	5 (9.6%)	0.98	1.01	0.25–4.02
Other	19 (45.2%)	27 (51.9%)	0.51	1.31	0.57–2.95

BMI, body mass index; ICU, intensive care unit. * No ** Yes *** only available for 2 × 2.

**Table 3 viruses-17-01622-t003:** Comparison between HCWs without and with only one SARS-CoV-2 infection during follow-up.

Variable	SARS-CoV-2 Infection During Follow-Up				
	No N = 53	Yes N = 28	*p*-Value	OR	CI 95%
Female gender n (%)	45 (84.9%)	25 (89.3%)	0.58	0.675	0.16–2.77
Age (mean ± SD, years)	52.2 ± 9.5	48.6 ± 11.1	0.36	0.965	0.921–1.011
Ethnicity n (%)					
White	18 (34%)	17 (60.7%)	**0.02**	0.333	0.129–0.859
Non-white	35 (66%)	11 (39.3%)			
Previous COVID-19 infection n (%)	33 (62.3%)	15 (53.6%)	0.44	0.699	0.27–1.76
Education n (%)					
Technician/Graduate	47 (88.7%)	13 (46.4%)	**<0.001**	9.038	2.924–27.93
Specialization/Master/Doctorate	6 (11.3%)	15 (53.6%)			
Hospital department n (%)					
ICU/emergency	16 (30.2%)	14 (50%)	**0.04**	0.269	0.073–0.991
Ward	17 (32.1%)	4 (14.3%)			
Laboratory/Radiology	7 (13.2%)	3 (10.7%)			

ICU, intensive care unit.

**Table 4 viruses-17-01622-t004:** Comparison of asymptomatic infections between the first and second detection of SARS-CoV-2 during follow-up.

Infection	Asymptomatic (n, %)	*p*-Value
First (n = 41)	10 (24.4)	0.001
Second (n = 13)	12 (92.3)	

## Data Availability

Data from this study will be made available upon reasonable request to the corresponding author.

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
