# Peer review of "Sociodemographic Associations and COVID-19 Symptoms Following One Year of Molecular Screening for SARS-CoV-2 Among Healthcare Workers"

_viruses, 2025, doi:10.3390/v17121622_

Round 1

Reviewer 1 Report

Comments and Suggestions for Authors

Sociodemographic Associations and COVID-19 Symptoms Following One Year of Molecular Screening for SARS-CoV-2 Among Healthcare Workers.

Review

The manuscript studies socio-demographic associations and COVID-19 symptoms toms following molecular screening for SARS-CoV-2 among healthcare workers during one year in Spain. This issue has interest from a public health perspective. Some suggestions could be proposed in order to improve the manuscript.

  1. In the abstract, when and where the study was performed could be described for a better understanding of readers.
  2. The participation rate of the study could be reported, considering that only the number of participants is mentioned and not the total of invited subjects to participate.
  3. The authors defined the study as a cross-sectional study. However, the same participants had follow-up for one year. A suggested definition may be prospective cohort study.
  4. Voluntary participation of different hospital’s departments could produce a selection bias.
  5. The name of the statistical program used could be mentioned.
  6. Table 1. A more detail of the exposure variables of SARS-CoV-2 of participants could be important to a better evaluation of the results.
  7. In Table 2, odds ratios (OR) were used, but ORs may be mentioned in the Material and Methods section.
  8. In Table 2, no ORs for continuous variable were estimated. However, they may be calculated by logistic or Poisson regressions.
  9. In Table 3, 81 participants had results of follow-up. However, the authors mention 94 participants with 15 participants lost, and this makes 79 participants. Could the authors clarify this point?
  10. The authors mention 28 new infections (Page 5, line 146). However, in Table 4, 54 infections were reported “between first and second detection of SARS-CoV-2 during the follow-up. “ Could the authors explain this situation?
  11. Only percent of new SARS-CoV-2 infections are reported. However, the incidence of new SARS-CoV-2 infections needs to know the persons at risk.
  12. No adjusted approach for potential confounding factors has been made, and a bias from confounding could occur.
  13. A statistical analysis by multivariable methods such as logistic regression or Poisson regression is recommended in order to adjustments of variables.
  14. Limitations of this study were not mentioned. However, some limitations could be indicated, including no control of confounding factors, no estimation of incidence rates of new SARS-CoV-2 infections, and a possible selection bias.
  15. The authors should follow the guidelines of the journal for the references.

Author Response

Reviewer: 1

Comment # 1: In the abstract, when and where the study was performed could be described for a better understanding of readers.

Reply # 1: The abstract was updated in the revised version of the manuscript, in order to include the period and the place the study was conducted. We would like to thank the reviewer for the opportunity to improve this section of the paper.

Comment # 2: The participation rate of the study could be reported, considering that only the number of participants is mentioned and not the total of invited subjects to participate.

Reply # 2: In total, 1,820 healthcare workers were estimated to be invited to participate in the study (either by direct contact or through common areas and website advertising). This information was added in the revised version of the manuscript (page 2, line 59). In addition, this limitation of HCWs participation was included in the added Study Limitations section of the manuscript (page 10, lines 334 to 347).

Comment # 3: The authors defined the study as a cross-sectional study. However, the same participants had follow-up for one year. A suggested definition may be prospective cohort study.

Reply # 3: Thank you for your comment. We corrected the description of the study in the revised manuscript, to prospective cohort study in Materials and Methods section (page 2, line 55).

Comment # 4: Voluntary participation of different hospital’s departments could produce a selection bias.

Reply # 4: We added this acknowledgment in Study Limitations section, which was added of the revised version of the paper (page 9, line 340).

Comment # 5: The name of the statistical program used could be mentioned.

Reply # 5: The data were analyzed using the statistical software Jamovi 2.7.9 and R 4.5.1. This information was added in Statistical Analysis section of the revised manuscript (page 3, line 115).

Comment # 6: Table 1. A more detail of the exposure variables of SARS-CoV-2 of participants could be important to a better evaluation of the results.

Reply # 6: As an exposure variable, we have added the number of household contacts of the participating HCWs (2.9 ± 1.3), in Table 1 of the revised version of the manuscript.

Comment # 7: In Table 2, odds ratios (OR) were used, but ORs may be mentioned in the Material and Methods section.

Reply # 7: We have mentioned the employment of odds rations (OR), in the Material and Methods section of the revised version of the manuscript (page 3, line 112).

Comment # 8: In Table 2, no ORs for continuous variable were estimated. However, they may be calculated by logistic or Poisson regressions.

Reply # 8: We have added OR for continuous variables in Table 2, in the revised version of the manuscript (page 5).

Comment # 9:  In Table 3, 81 participants had results of follow-up. However, the authors mention 94 participants with 15 participants lost, and this makes 79 participants. Could the authors clarify this point.

Reply # 9: We appreciate the comment and would like to correct this number. During the study, thirteen healthcare workers withdrew participation. Accordingly, 81 participants had results of follow-up. This number was corrected in the revised version of the manuscript (page 3, line 120).

Comment # 10: The authors mention 28 new infections (Page 5, line 146). However, in Table 4, 54 infections were reported “between first and second detection of SARS-CoV-2 during the follow-up. Could the authors explain this situation?

Reply # 10: During the study, a total of fifty-four SARS-CoV-2 infections in 41 participants were detected, as can be inferred from data presented in Table 4; n=41 (first infection) + n=13 (second infection). Consequently, 28 participants had only one infection (=41-13). Indeed, Table 4 was made to compare symptom occurrence between first and second infections. However, on Table 3, we compare sociodemographic data between individuals without infection throughout the study (n=53) with participants with only one infection (n= 28). We chose not to include individuals with two infections in this analysis, since their characteristics could differ from those with one infection. Additionally, participant number would not aloud comparisons between individuals with one and two infections. In order to make clear this issue, we updated the text in item 3.3 to “only one infection” (lines 185 and 196).

Comment # 11: Only percent of new SARS-CoV-2 infections are reported. However, the incidence of new SARS-CoV-2 infections needs to know the persons at risk.

Reply # 11: In the revised version of the manuscript, we informed the number of potential participants who were invited to join the study (1,820 healthcare workers, at page 2, line 60). However, the reduced number of individuals who agreed to participate (n=94) do not necessarily represent the universe of workers at the hospital, as acknowledged in the Study Limitations of the revised version of the manuscript (page 9, line 341).

Comment # 12: No adjusted approach for potential confounding factors has been made, and a bias from confounding could occur.

Reply # 12: We performed binomial logistic regression, in order to evaluate the effects of a potential confounding factor (age), both for COVID-19 infection previous to study inclusion and during the study (supplementary Tables 1 and 2). However, no association with infection was found. These analyses were described in the revised version of the manuscript (page 6, line 192).

Comment # 13: A statistical analysis by multivariable methods such as logistic regression or Poisson regression is recommended in order to adjustments of variables.

Reply # 13: We appreciate the reviewer comment, and included logistic regression analysis, as described in the previous answer.

Comment # 14: Limitations of this study were not mentioned. However, some limitations could be indicated, including no control of confounding factors, no estimation of incidence rates of new SARS-CoV-2 infections, and a possible selection bias.

Reply # 14: Thank you for your comment. We added a Study Limitations section addressing these issues, in the revised version of the manuscript (page 9, lines 334 to 347).   

Comment # 15: The authors should follow the guidelines of the journal for the references.

Reply # 15: The journal guidelines for references were adopted in the revised version of the manuscript.  

Reviewer 2 Report

Comments and Suggestions for Authors

The manuscript addresses an important and timely topic regarding SARS-CoV-2 infection and reinfection among healthcare workers (HCWs). The longitudinal molecular screening approach provides valuable data on asymptomatic infection dynamics and occupational risk factors. However, several issues concerning methodological clarity, data presentation, and interpretation should be addressed to strengthen the paper’s rigor and readability.

Major Points

  1. Study Design and Cohort Definition

    • The study is described as cross-sectional, yet it includes longitudinal follow-up and repeated RT-PCR testing. Please clarify whether the design is prospective cohort rather than cross-sectional, and adjust terminology accordingly throughout the text.

  2. Statistical Analysis

    • Statistical methods need more explanations. Please specify which tests were used for each comparison, whether assumptions were verified (e.g., normality), and how missing data or participant dropouts were handled.

    • Indicate if multiple testing correction was applied to control for Type I error.

    • Provide confidence intervals for key estimates (infection rates, ORs) to allow assessment of precision.

  3. Sample Size and Power

    • Discuss whether a sample size or power calculation was conducted before recruitment to justify the number of participants.

  4. Interpretation of Sociodemographic Associations

    • The finding that white HCWs had higher infection rates during follow-up than non-white individuals needs a more cautious interpretation. Explore possible confounding factors such as occupational roles, exposure levels, or differences in reporting rather than implying causality.

  5. Temporal Trends and Variants

    • The three infection peaks are mentioned but not linked to known epidemiological waves or variant circulation in Brazil. Please provide contextual data (e.g., national variant prevalence or public health measures at those times) to support interpretation.

  6. Vaccination Data

    • The study notes an average of 4.2 vaccine doses per participant but does not specify vaccine types or time since last dose. These details are critical for interpreting reinfection dynamics.

  7. Symptom Definition and Data Collection

    • Clarify how symptoms were assessed (self-report, checklist, or medical evaluation) and over what timeframe after a positive test. The increased asymptomatic rate may reflect underreporting or recall bias.

  8. Figure 1

    • Figure 1 should include a clear time scale, labels for departments, and indication of total cases per wave. The figure caption should explain abbreviations (e.g., “Department A”).

  9. Ethical and Data Availability Statements

    • The statements are appropriate, but it would be useful to clarify how confidentiality and biosafety procedures were maintained during repeated sampling.

  10. Discussion and Conclusions

    • The discussion repeats descriptive results rather than offering deeper analysis or comparison to global data on HCW infection rates post-vaccination.

    • The conclusion should be more concise and directly supported by the study findings—avoid overly general policy recommendations not directly examined in the data.

Minor Points

  1. Correct typographical errors throughout (e.g., “Boby mass index” → “Body mass index”; “Dinamics” → “Dynamics”).

  2. Some tables are missing units and need consistent formatting (e.g., Table 1 and Table 3).

  3. Replace “NaN*** only available for 2x2” with a proper footnote or remove if not applicable.

  4. The reference style should be standardized according to Viruses journal guidelines.

  5. Avoid repetition in the Author Contributions section (“writing-original draft preparation” appears twice).

  6. Ensure all abbreviations are defined at first mention (e.g., OHS, ICU).

  7. Consider moving detailed PCR assay information to Supplementary Material.

Author Response

Reviewer: 2

Major points

Comment # 1: Study Design and Cohort Definition: The study is described as cross-sectional, yet it includes longitudinal follow-up and repeated RT-PCR testing. Please clarify whether the design is prospective cohort rather than cross-sectional, and adjust terminology accordingly throughout the text.

Reply # 1: We corrected the description of the study in the revised version of the manuscript to prospective cohort study, in Materials and Methods section (page 2, line 55).

Comment # 2: Statistical Analysis: Statistical methods need more explanations. Please specify which tests were used for each comparison, whether assumptions were verified (e.g., normality), and how missing data or participant dropouts were handled.

Reply # 2: The revised version of the manuscript included a description of the test employed for comparisons and the normality test (page 3, line 109, and lines 113-115). Patients with missing data were excluded from analysis (page 3, line 116). During the study, 13 HCWs withdrew from participation. However, their regular testing data was kept, until their leave from the study. This information was added in page 3, line 122.

Comment # 3: Statistical Analysis: Indicate if multiple testing correction was applied to control for Type I error.

Reply # 3: We performed binomial logistic regression, in order to test the effects of a potential confounding factor (age), both for COVID-19 infection previous to study inclusion and during the study (supplementary Tables 1 and 2). However, no association with the infection was found, a finding possibly associated with sample size. These analyses were described in the revised version of the manuscript (page 4, line 190).

Comment # 4: Statistical Analysis: Provide confidence intervals for key estimates (infection rates, ORs) to allow assessment of precision.

Reply # 4: We have added ORs for continuous variables in Table 2, in the revised version of the manuscript.

Comment # 5: Sample Size and Power: Discuss whether a sample size or power calculation was conducted before recruitment to justify the number of participants

Reply # 5: In the revised version of the manuscript, we informed the number of potential participants who were invited to join the study (1,820 HCWs, page 2, line 60), as a convenience sampling. However, a reduced number agreed to participate in the research protocol (n=94). We recognize that this sample do not necessarily represent the universe of HCWs at the hospital, as acknowledged in the Study Limitations section (page 9, line 341).

Comment # 6: Interpretation of Sociodemographic Associations: The finding that white HCWs had higher infection rates during follow-up than non-white individuals need a more cautious interpretation. Explore possible confounding factors such as occupational roles, exposure levels, or differences in reporting rather than implying causality

Reply # 6: We performed binomial logistic regression, in order to test the effects of a potential confounding factor, both for COVID-19 infection previous to study inclusion and during the study (supplementary Tables 1 and 2). However, no association was found, a finding possibly associated with sample size. The concept of race being associated with socioeconomic conditions, such as access to higher education in the healthcare field, is emphasized in the Discussion section of the revised version of the manuscript (page 8, lines 296-299).

Comment # 7: Temporal Trends and Variants: The three infection peaks are mentioned but not linked to known epidemiological waves or variant circulation in Brazil. Please provide contextual data (e.g., national variant prevalence or public health measures at those times) to support interpretation

Reply # 7: The authors would like to thank the reviewer for the comment. We sent swab material from the study participants for genomic sequencing, which appeared to show an association with the emergence of SARS-CoV-2 variants in the city of Rio de Janeiro and in Brazil, during 2022 and 2023. Genomic sequencing revealed the presence of distinct lineages of SARS-CoV-2, all belonging to the Omicron group. Lineages detected in the study included BA.5.2.1, BQ.1.1, DL.1, and GK.1.1. However, a very small number of samples displayed suitable DNA integrity for the evaluation (n=7). Accordingly, the authors understood that this analysis would be of limited interpretation and was beyond the scope of the present manuscript.

Comment # 8: Vaccination Data: The study notes an average of 4.2 vaccine doses per participant but does not specify vaccine types or time since last dose. These details are critical for interpreting reinfection dynamics.

Reply # 8: The vaccines available in Brazil during this period were Coronavac®, AstraZenica®, Janssen®, Pfizer®, bivalent Pfizer® and Sinovac®. The time interval between the last dose of the COVID-19 vaccine and the first infection in asymptomatic individuals was 168.8± 71.9 days and in symptomatic individuals, 168.5±107.5 days. This information was added in the revised version of the manuscript (page 3, lines 123-127).

Comment # 9: Symptom Definition and Data Collection:  Clarify how symptoms were assessed (self-report, checklist, or medical evaluation) and over what timeframe after a positive test. The increased asymptomatic rate may reflect underreporting or recall bias.

Reply # 9: In the day of each swab collection, the project's nursing team performed an anamnesis directed to symptoms related to COVID-19. Symptomatic HCWs were not followed up until the next test collection. This information was added to the revised version of the manuscript (page 2, line 69).

Comment # 10: Figure 1: Figure 1 should include a clear time scale, labels for departments, and indication of total cases per wave. The figure caption should explain abbreviations (e.g., “Department A”).

Reply # 10: The manuscript Figure 1 was revised with the aid of MDPI professional services, in order to improve presentation.

Comment # 11: Ethical and Data Availability Statements: The statements are appropriate, but it would be useful to clarify how confidentiality and biosafety procedures were maintained during repeated sampling.

Reply # 11: The Informed Consent Form (ICF) was administered at the first visit. This information was added in the revised version of the manuscript (page 2, line 63). Biosafety procedures were further detailed in page 3, lines 83-85

Comment # 12: Discussion and Conclusions: The discussion repeats descriptive results rather than offering deeper analysis or comparison to global data on HCW infection rates post-vaccination.

Reply # 12: The manuscript Discussion section was revised and descriptive results were suppressed. Additional analysis and comparisons to global data on HCW infection rates post-vaccination were included (page 8, line 264 and lines 268 to 271).

Comment # 13: Discussion and Conclusions: The conclusion should be more concise and directly supported by the study findings—avoid overly general policy recommendations not directly examined in the data.

Reply # 13: The manuscript Discussion and Conclusions sections were revised in order to improve adequacy to the study findings and avoid generally policy recommendations.

Minor points

Comment # 14: Correct typographical errors throughout (e.g., “Boby mass index” → “Body mass index”; “Dinamics” → “Dynamics”)..

Reply # 14: Thank you for your comment. The typographical errors were corrected with the aid of MDPI professional services.

Comment # 15: Some tables are missing units and need consistent formatting (e.g., Table 1 and Table 3).

Reply # 15: The Tables formatting and data were revised in the current version of the manuscript.

Comment # 16: Replace “NaN*** only available for 2x2” with a proper footnote or remove if not applicable.

Reply # 16: We replaced the term by a footnote in Tables of the revised version of the manuscript.

Comment # 17: The reference style should be standardized according to Viruses journal guidelines.

Reply # 17: The reference style was revised according to Viruses journal guidelines in the current version of the manuscript.

Comment # 18: Avoid repetition in the Author Contributions section (“writing-original draft preparation” appears twice).

Reply # 18: This issue was corrected in the revised version of the manuscript.

Comment # 19: Ensure all abbreviations are defined at first mention (e.g., OHS, ICU)..

Reply # 19: The definitions of all abbreviations were defined at first mention in the revised version of the manuscript.

Comment # 20: Consider moving detailed PCR assay information to Supplementary Material.

Reply # 20: The authors have chosen to direct the reader to an Institutional website (US FDA) for detailed information regarding PCR assay technique (available at: https://www.fda.gov/media/134922/download, at page3, line 94)

Round 2

Reviewer 1 Report

Comments and Suggestions for Authors

The authors have addressed all the suggestions of our review. The small sample of this study limits the generalizations of the results.

Author Response

Reviewer 1 had no remaining comments on Round 2 of the manuscript review process.

Reviewer 2 Report

Comments and Suggestions for Authors

The manuscript have been substantially improved since the last version. However I have some remarks:

Abstract: The abstract summarizes key findings but could be improved by  stating the study design (“prospective cohort study of 94 HCWs with biweekly RT-PCR screening”). It mentions statistical significance (p values) but lacks effect sizes or confidence intervals, which would add depth. Interpretation could be more concise and linked more clearly to specific results

Introduction: The introduction is well contextualized but could benefit from a learer identification of the knowledge gap (Why is longitudinal molecular screening in HCWs underexplored?) and clarification of how this study builds on prior research (e.g., SIREN study and similar cohorts)

Methods: (a) study design  section:  provide justification for biweekly PCR screening frequency, (b) aymptom assessment: (i) please make clear whether symptoms were self-reported or assessed by a clinician, (ii) please explain how asymptomatic cases were differentiated from presymptomatic cases, (c) Statistical methods: (i) some statistical details are vague (e.g., which variables were included in logistic regression?), please clarify the details, (ii) the use of OR and CI should be clarified for adjusted vs unadjusted analyses, (iii) note on handling missing data is brief - specify extent and method.

Results: Tables are informative, but: (i) Table 1 lacks p values to indicate whether any differences are meaningful, (ii) In Table 3, the strong association between education level and infection risk is intriguing—this deserves further discussion in context of confounding, (iii) Figure 1 is mentioned but not clearly described (e.g., what do “Department A/B/C” correspond to?), (iv) The presentation of percentages and sample sizes should be consistent throughout. The results section mixes analysis and interpretation; keep interpretation mainly in the Discussion.

Discussion:  (a) Some key limitations are briefly mentioned but not well elaborated (e.g., selection bias, small sample, heterogeneity in exposure), (b) there is no mention of potential confounders such as PPE use or adherence, variant identification, or serological data, (c) The explanation for why white participants showed higher infection rates during follow-up should be expanded or qualified more cautiously

Minor comments: (a) Typographical inconsistencies (spacing, punctuation, mixed use of commas/periods in numbers), (b) Repeated use of “reinfected,” but unclear whether confirmed reinfection was defined by genomic sequencing or only PCR positivity, (c) Some long paragraphs can be split (d) consider adding a visual timeline or flowchart (Participant tracking over 576-day follow-up).

Author Response

Reviewer: 2

Comment # 1: Abstract: The abstract summarizes key findings but could be improved by stating the study design (“prospective cohort study of 94 HCWs with biweekly RT-PCR screening”). It mentions statistical significance (p values) but lacks effect sizes or confidence intervals, which would add depth. Interpretation could be more concise and linked more clearly to specific results.

Reply # 1: The authors would like to thank the opportunity to further improve our manuscript. The abstract was updated in order to state the number of participants, in the revised version of the manuscript. In addition, confidence intervals were added (both in abstract and in Table 3, page 10), and the abstract interpretation comments were edited in order to relate more concisely to our results.

Comment # 2: Introduction: The introduction is well contextualized but could benefit from a clearer identification of the knowledge gap (Why is longitudinal molecular screening in HCWs underexplored?) and clarification of how this study builds on prior research (e.g., SIREN study and similar cohorts).

Reply # 2: The manuscript Introduction section was revised, in order to acknowledge previous studies and particularly the key SIREN study (page 2, lines 46-51), and to highlight the contribution of the present study (page 2, lines 48-54).

Comment # 3: Methods: (a) study design  section:  provide justification for biweekly PCR screening frequency, (b) symptom assessment: (i) please make clear whether symptoms were self-reported or assessed by a clinician, (ii) please explain how asymptomatic cases were differentiated from presymptomatic cases, (c) Statistical methods: (i) some statistical details are vague (e.g., which variables were included in logistic regression?), please clarify the details, (ii) the use of OR and CI should be clarified for adjusted vs unadjusted analyses, (iii) note on handling missing data is brief - specify extent and method.

Reply # 3:

(a) In the revised version of the manuscript, the biweekly PCR screening interval was justified based on prior literature assessing the duration of SARS-CoV-2 shedding in biological specimens, which generally spans from approximately four days before to 17 days after the onset of symptoms. (page 3, lines 85-88).

(b, i) Symptom assessment was based on participant recall, consistent with the approach employed by previous studies. This explanation was added in the revised version of the manuscript (page 2, lines 79, 80).

(b, ii) In all instances of asymptomatic positive participants throughout the study, no symptom was self-reported by the next biweekly testing. This observation was added to the revised manuscript (page 7, lines 232,233).

(c, i) The sample size constrained the analysis of variables using logistic regression, which in this study was restricted to participant age (the variable included in regression was informed in Methods section, page 4, lines 130,131).  This limitation was acknowledged in the revised Limitations section (page 10, lines 348-350).

(c, ii) OR and CI values were incorporated into the revised manuscript tables where applicable (Tables 2 and 3, and Supplementary Tables 1 and 2).

(c, iii) Participants with missing data were designated for exclusion from the analysis; however, no such cases occurred in this study. This information was added in the revised Methods section of the manuscript (page 4, lines 132,133).

Comment # 4: Results: Tables are informative, but: (i) Table 1 lacks p values to indicate whether any differences are meaningful, (ii) In Table 3, the strong association between education level and infection risk is intriguing—this deserves further discussion in context of confounding, (iii) Figure 1 is mentioned but not clearly described (e.g., what do “Department A/B/C” correspond to?), (iv) The presentation of percentages and sample sizes should be consistent throughout. The results section mixes analysis and interpretation; keep interpretation mainly in the Discussion.

Reply # 4:

(i) The manuscript Table 1 presents standard descriptive statistics intended to characterize the sociodemographic profile of the study participants. Accordingly, p-values were not included, as no comparative analyses were conducted.

(ii) As acknowledged in the previous revised version of the manuscript in the Discussion section, worldwide, gender and ethnicity are associated to the educational level in healthcare workforce, with higher proportion of women and non-white individuals historically working in lower-education activities. Accordingly, the effects of ethnicity or educational levels per se in SARS-CoV-2 infections, cannot be separated in our study. HCWs with different educational levels may perform different functions within the health institution, which may influence their exposure to the virus. HCWs with lower levels of education may be more frequently involved in operational functions that require greater physical contact with patients, increasing the risk of infection (page 9, lines 308-315). The authors believe this point clarifies that any potential effect of ethnicity should be interpreted within a broader socioeconomic context, extending beyond considerations of race alone.

(iii) Following an internal discussion among the study authors, letter coding was used to avoid identifying hospital departments that experienced peaks of HCW infections during the pandemic. This approach nevertheless enabled the observation of synchrony in infection patterns across departments, as highlighted in the manuscript. (page 7, lines 246-247).

(iv) The presentation of percentages and sample sizes was revised in order to improve consistency, with the correction in Table 3 of the revised manuscript (page 6). Data interpretation comments were removed in Results of the revised manuscript version.

Comment # 5: Discussion: (a) Some key limitations are briefly mentioned but not well elaborated (e.g., selection bias, small sample, heterogeneity in exposure), (b) there is no mention of potential confounders such as PPE use or adherence, variant identification, or serological data, (c) The explanation for why white participants showed higher infection rates during follow-up should be expanded or qualified more cautiously

Reply # 5:

(a) Additional Study Limitations comments were included in the revised version of the manuscript, regarding selection bias (page 10, lines 348,349; lines 352-354 and lines 356-361).

(b) The authors would like to thank the reviewer for the comment. The absence of data regarding variant identification or participant serological data were included in the revised version of the manuscript (page 10, lines 359-361). Concerning viral variant identification in our study, we sent swab material from participants for genomic sequencing. The analyses revealed the presence of distinct lineages of SARS-CoV-2, all belonging to the Omicron group. The lineages detected in the study included BA.5.2.1, BQ.1.1, DL.1, and GK.1.1. However, a very small number of samples displayed suitable DNA integrity for the evaluation (n=7). Accordingly, the authors understood that this analysis would be of limited interpretation and was beyond the scope of the present manuscript.

(c) The authors acknowledge that interpreting an association between infection rates and ethnicity alone requires caution and must be framed within an appropriate socioeconomic context. Accordingly, in the Discussion section of the previously revised manuscript, we added commentary addressing how ethnicity and gender relate to educational level, professional role, and occupational exposure risk (page 9, lines 308-316).

Comment # 6: Minor comments: (a) Typographical inconsistencies (spacing, punctuation, mixed use of commas/periods in numbers), (b) Repeated use of “reinfected,” but unclear whether confirmed reinfection was defined by genomic sequencing or only PCR positivity, (c) Some long paragraphs can be split (d) consider adding a visual timeline or flowchart (Participant tracking over 576-day follow-up).

Reply # 6:

(a) Typographical inconsistencies were corrected in the current revised version of the manuscript, in addition to the previous review from MDPI professional editing services.

(b) The definition of reinfection during the study was added in the revised manuscript version (page 3, lines 110-113).

(c) In the revised version of the manuscript, long paragraphs were divided for improved clarity and readability. (page 8, lines 275-278 and page 9, lines 333-336).

(d) As noted in the Data Availability Statement, additional data from this study will be made available upon reasonable request to the corresponding author. The study conducted an average of 22.7 ± 11.1 tests per participant, totaling more than 2,100 assays, and these data can be provided upon request.